# Effect of short inter-pregnancy interval on perinatal and maternal outcomes among pregnant women in SSA 2023: Systematic review and meta-analysis

**Fentahun Yenealem Beyene**⊙*, **Kihinetu Gelaye Wudineh, Simachew Animen Bantie, Azimeraw Arega Tesfu**

Department of Midwifery, College of Medicine and Health Sciences, Bahir Dar University, Bahir Dar, Ethiopia

* yenefenta84@gmail.com

## Abstract

### Background

After a live birth, the recommended interval before attempting the next pregnancy is at least 24 months (birth to pregnancy interval) in order to reduce the risk of adverse maternal, perinatal and infant outcomes. Short inter pregnancy interval associated with adverse perinatal and maternal outcomes.

### Objective

The objective of this review was to determine the effect of short inter pregnancy interval on perinatal and maternal outcomes in Sub-Saharan Africa 2023.

### Methods

A systematic and a comprehensive literature searching mechanism were used without any restriction, through Google scholar, PubMed, Scopus, Web of Sciences, and Grey literature databases for reporting the effect of short inter pregnancy interval. The JBI approach to critical appraisal, study selection, data extraction, and data synthesis was used for this review. All statistical analyses were done using STATA version17 software for windows, and meta-analysis was used with a random-effects method. The results are presented using texts, tables and forest plots with measures of effect and 95% confidence interval.

### Results

Thirteen studies were included in this review and most of the studies level of heterogeneity across the study was considerable, mainly due to methodological variations, Statistical heterogeneity, and population and intervention variations of included studies. The effect of short inter pregnancy interval on perinatal and maternal outcome were low birth weight(RR (RR (95% CI) 1.98 (1.48, 2.47); $I^2$:62.97%, preterm birth (RR (95% CI) 1.67 (1.31, 2.03); $I^2$:51%, intra uterine growth retardation(RR (95% CI) 3.78 (2.07, 5.49); $I^2$: 8.52%, low

**Data Availability Statement:** All relevant data are within the manuscript and its Supporting information files.

**Funding:** The author(s) received no specific funding for this work.

**Competing interests:** The authors have declared that no competing interests exist.

**Abbreviations:** APGAR, Appearance, Pulse, Grimace, Activity and Respiration; CI, Confidence Interval; IUGR, Intra Uterine Growth Retardation; NGO, Non-Governmental Organization; OR, Odds Ratio; PRISMA, Preferred Reporting Items for Systematic Reviews and Meta-Analyses; PROM, Premature Rupture of Membrane; PTB, Preterm Birth; SIPI, Short Inter Pregnancy Interval; SSA, Sub Saharan Africa; WHO, World Health Organization.

APGAR score(RR (95% CI) 3.49 (1.41, 5.57)); $I^2$: 71.11%, premature rapture of membrane (RR (95% CI) 2.87 (1.22, 4.51)); $I^2$: 49.22%, perinatal mortality(RR (95% CI) 2.95 (1.10, 4.81)); $I^2$: 54.37% and maternal anemia(RR (95% CI) 3.06 (2.12, 3.99)); $I^2$: 74.74%.

## Conclusions

As per our review the main effect of short inter pregnancy interval is low birth weight, preterm birth, intra uterine growth retardation, low APGAR score, premature rupture of membrane, perinatal mortality and maternal anemia. This might be very useful for healthcare policy-makers and NGOs to emphasize on it.

## Introduction

### Background

Short inter pregnancy is defined as a period of less than 24 months between a live birth and the following conception. It is advised to wait at least 24 months after a live birth before trying to conceive again in order to lower the risk of unfavorable maternal, perinatal, and baby outcomes [1]. Worldwide studies have shown that poorly spaced pregnancies have detrimental effects on both mother and child health [2–4]. Each year, there are over 303,000 maternal deaths, 2.7 million infant deaths, and 2.6 million stillbirths worldwide. Around 99% of deaths among those happened in underdeveloped nations [5, 6]. Sub-Saharan Africa is one of the areas having a high prevalence of fatalities among mothers and newborns. It is responsible for more than half of all maternal and newborn deaths that occur globally [6, 7]. Evidence from systematic reviews and meta-analyses shows that short intervals between pregnancies are independently associated with a higher risk of adverse maternal, perinatal, baby, and child outcomes [8–10]. Interpregnancy intervals shorter than 24 months are significantly associated with increased risk of adverse perinatal outcomes such as preterm birth, low birth-weight, and small for gestational age [8, 11, 12]; premature membrane rupture [10, 13, 14]; Abruptio placentae and placenta previa [8, 15]; uterine rupture in women attempting a vaginal birth after previous cesarean delivery [8, 16, 17]; low APGAR score [18, 19]; perinatal and neonatal mortality [10].

It has been hotly debated how short gaps between pregnancies could impact maternal, perinatal, baby, and child health [20–22]. The typical biological or behavioral orientations adopted by hypotheses are; the Maternal Depletion Syndrome [23], the Folate Depletion Theory [24, 25], and putative nutritional-related causative mechanisms [26].

Generally short birth interval has effects on socio-economic and the reproductive behaviors of individual in related to health status of the child bearing mother and their children. In developing country including SSA, maternal and child complication associated with short birth interval practice remain highly significant. There hasn't been a comprehensive study of the effect of short-inter pregnancy interval on perinatal outcome at larger level, only small-scale research in different regional and zonal level. Therefore, the purpose of this meta-analysis is to estimate the effect of short-inter pregnancy interval on perinatal outcome at continent level in a more comprehensive manner. The results of this study would emphasize the significance and urgency of expanding the prevention modalities of short inter pregnancy interval to minimalize its negative perinatal effect. Understanding the effect of short-inter pregnancy interval in SSA may also help determine the best intervention to use in order to lessen the severity of the issue, enhance mother and child health, and end the burden of SIPI in SSA. As a result, we

conducted a systematic review and meta-analysis to assess the effect of short-inter pregnancy interval on perinatal outcome in SSA.

### Review question

The question/s of this review is: what is the effect of short-inter pregnancy interval on perinatal outcome?

**Key words.** Birth interval complications; Birth spacing; Birth to birth interval; Birth to Conception; pregnant ladies.

### Inclusion criteria

**Participants.** This review was done on pregnant women with at least one previous history of one a live birth. The review included studies conducted in SSA countries; which are classified as 5 central Africa; 8 East Africa; 10 South Africa and 18 West Africa) [27].

### Intervention

This review considered pregnant women with short inter pregnancy interval (inter pregnancy interval < 24 months).

### Comparator

Pregnant women with optimal inter pregnancy interval ((inter pregnancy interval 2–5 year) were included.

### Outcomes

This review considered studies that included perinatal outcomes including; low birth weight, preterm birth, IUGR, low APGAR score, perinatal mortality, PROM and maternal anemia.

### Types of studies

This review considered cohort studies (both prospective and retrospective), case control studies and analytical cross-sectional studies with only English language textual were included.

## Methods

This systematic review was conducted in accordance with JBI methodology for Systematic reviews of etiology and risk [28]. We used the Preferred Reporting Items for Systematic Reviews and Meta-Analyses (PRISMA) criteria to review and present the findings of this systematic review and meta-analysis with (PROSPERO registration number CRD42023407644) [29] (Table 1).

### Search strategy

The search strategy was aimed to find both published and unpublished studies. A two–step search strategy was considered in this review. First an initial limited search of MEDLINE (PubMed) and Google scholar was undertaken to identify articles on the topic. The text words contained in the titles and abstracts of relevant articles, and the index terms used to describe the articles were used to develop a full search strategy for report the name of the relevant databases/information sources (Table 2). The second search strategy was, including all identified keywords and index terms, was taken for each included database and/or information source. The reference list of all included sources of evidence was screened for additional studies. *The*

**Table 1. PRISMA 2020 checklist.**

| Section and Topic | Item # | Checklist item | Location where item is reported |
|---|---|---|---|
| **TITLE** | | | I |
| Title | 1 | Identify the report as a systematic review. | I |
| **ABSTRACT** | | | II |
| Abstract | 2 | See the PRISMA 2020 for Abstracts checklist. | II |
| **INTRODUCTION** | | | 1 |
| Rationale | 3 | Describe the rationale for the review in the context of existing knowledge. | 1–2 |
| Objectives | 4 | Provide an explicit statement of the objective(s) or question(s) the review addresses. | 2 |
| **METHODS** | | | 3 |
| Eligibility criteria | 5 | Specify the inclusion and exclusion criteria for the review and how studies were grouped for the syntheses. | 2 |
| Information sources | 6 | Specify all databases, registers, websites, organisations, reference lists and other sources searched or consulted to identify studies. Specify the date when each source was last searched or consulted. | 3 |
| Search strategy | 7 | Present the full search strategies for all databases, registers and websites, including any filters and limits used. | 3 |
| Selection process | 8 | Specify the methods used to decide whether a study met the inclusion criteria of the review, including how many reviewers screened each record and each report retrieved, whether they worked independently, and if applicable, details of automation tools used in the process. | 3 |
| Data collection process | 9 | Specify the methods used to collect data from reports, including how many reviewers collected data from each report, whether they worked independently, any processes for obtaining or confirming data from study investigators, and if applicable, details of automation tools used in the process. | 3–4 |
| Data items | 10a | List and define all outcomes for which data were sought. Specify whether all results that were compatible with each outcome domain in each study were sought (e.g. for all measures, time points, analyses), and if not, the methods used to decide which results to collect. | 2 |
| | 10b | List and define all other variables for which data were sought (e.g. participant and intervention characteristics, funding sources). Describe any assumptions made about any missing or unclear information. | 2 |
| Study risk of bias assessment | 11 | Specify the methods used to assess risk of bias in the included studies, including details of the tool(s) used, how many reviewers assessed each study and whether they worked independently, and if applicable, details of automation tools used in the process. | 4 |
| Effect measures | 12 | Specify for each outcome the effect measure(s) (e.g. risk ratio, mean difference) used in the synthesis or presentation of results. | 5 |
| Synthesis methods | 13a | Describe the processes used to decide which studies were eligible for each synthesis (e.g. tabulating the study intervention characteristics and comparing against the planned groups for each synthesis (item #5)). | 4 |
| | 13b | Describe any methods required to prepare the data for presentation or synthesis, such as handling of missing summary statistics, or data conversions. | 4 |
| | 13c | Describe any methods used to tabulate or visually display results of individual studies and syntheses. | 4 |
| | 13d | Describe any methods used to synthesize results and provide a rationale for the choice(s). If meta-analysis was performed, describe the model(s), method(s) to identify the presence and extent of statistical heterogeneity, and software package(s) used. | 5 |
| | 13e | Describe any methods used to explore possible causes of heterogeneity among study results (e.g. subgroup analysis, meta-regression). | 5 |
| | 13f | Describe any sensitivity analyses conducted to assess robustness of the synthesized results. | 5 |
| Reporting bias assessment | 14 | Describe any methods used to assess risk of bias due to missing results in a synthesis (arising from reporting biases). | 5 |
| Certainty assessment | 15 | Describe any methods used to assess certainty (or confidence) in the body of evidence for an outcome. | 4 |
| **RESULTS** | | | 5 |
| Study selection | 16a | Describe the results of the search and selection process, from the number of records identified in the search to the number of studies included in the review, ideally using a flow diagram. | 5 |
| | 16b | Cite studies that might appear to meet the inclusion criteria, but which were excluded, and explain why they were excluded. | 5 |

*(Continued)*

**Table 1.** (Continued)

| Section and Topic | Item # | Checklist item | Location where item is reported |
|---|---|---|---|
| Study characteristics | 17 | Cite each included study and present its characteristics. | 5 |
| Risk of bias in studies | 18 | Present assessments of risk of bias for each included study. | 6–7 |
| Results of individual studies | 19 | For all outcomes, present, for each study: (a) summary statistics for each group (where appropriate) and (b) an effect estimate and its precision (e.g. confidence/credible interval), ideally using structured tables or plots. | 5–7 |
| Results of syntheses | 20a | For each synthesis, briefly summarise the characteristics and risk of bias among contributing studies. | 5–7 |
| | 20b | Present results of all statistical syntheses conducted. If meta-analysis was done, present for each the summary estimate and its precision (e.g. confidence/credible interval) and measures of statistical heterogeneity. If comparing groups, describe the direction of the effect. | 5–7 |
| | 20c | Present results of all investigations of possible causes of heterogeneity among study results. | 6–7 |
| | 20d | Present results of all sensitivity analyses conducted to assess the robustness of the synthesized results. | Not done |
| Reporting biases | 21 | Present assessments of risk of bias due to missing results (arising from reporting biases) for each synthesis assessed. | 6–7 |
| Certainty of evidence | 22 | Present assessments of certainty (or confidence) in the body of evidence for each outcome assessed. | 5–7 |
| **DISCUSSION** | | | |
| Discussion | 23a | Provide a general interpretation of the results in the context of other evidence. | 7 |
| | 23b | Discuss any limitations of the evidence included in the review. | 10 |
| | 23c | Discuss any limitations of the review processes used. | 10 |
| | 23d | Discuss implications of the results for practice, policy, and future research. | 11 |
| **OTHER INFORMATION** | | | |
| Registration and protocol | 24a | Provide registration information for the review, including register name and registration number, or state that the review was not registered. | II |
| | 24b | Indicate where the review protocol can be accessed, or state that a protocol was not prepared. | N/A |
| | 24c | Describe and explain any amendments to information provided at registration or in the protocol. | N/A |
| Support | 25 | Describe sources of financial or non-financial support for the review, and the role of the funders or sponsors in the review. | 12 |
| Competing interests | 26 | Declare any competing interests of review authors. | 12 |
| Availability of data, code and other materials | 27 | Report which of the following are publicly available and where they can be found: template data collection forms; data extracted from included studies; data used for all analyses; analytic code; any other materials used in the review. | 12 |

*From*: Page MJ, McKenzie JE, Bossuyt PM, Boutron I, Hoffmann TC, Mulrow CD, et al. The PRISMA 2020 statement: an updated guideline for reporting systematic reviews. BMJ 2021;372:n71. 10.1136/bmj.n71 For more information, visit: http://www.prisma-statement.org/

most recent search date was April 30, 2023, and studies with a publication up until April 2023 were considered for the review. Only articles written in English were included for review.

## Information sources

The full database search included PubMed, Scopus, CINAHL, Google scholar, Web of Science and ProQuest Dissertations and Theses Global database was searched for unpublished studies.

## Study selection

Following the search, all identified citations were organized and uploaded into EndNote X8 (Clarivate Analytics, PA, USA) and duplicates removed. Following a pilot test, titles and abstracts were screened by two or more independent reviewers for assessment against the inclusion criteria for the review. Potentially relevant studies retrieved in full and their citation

**Table 2. Data extraction of effect of short interpregnancy interval on perinatal and maternal outcome in SSA; 2023.**

| Data base | Searching terms | Searching date | Number of studies |
|---|---|---|---|
| PubMed | ((((((((((((((((Pregnant Women[MeSH Terms]) OR (Pregnancy[MeSH Terms])) OR (Pregnant Woman)) OR (Woman, Pregnant)) OR (Women, Pregnant)) OR (Pregnancies)) OR (Gestation)) OR (Gravidity)) OR (Pregnant lady)) OR (Pregnant girl)) OR (Conceived girl)) OR (Conceived lady)) OR (Conceived women)) AND (((((((((Short inter pregnancy interval) OR (Short inter birth interval)) OR (inter pregnancy interval)) OR (inter birth interval)) OR (short inter-conception period)) OR (short inter-conception interval)) OR (inter pregnancy interval <24 months)) OR (inter pregnancy interval <18 months)) OR (inter pregnancy interval <6months))) AND (((((((Perinatal out come) OR (Low birth weight[MeSH Terms])) OR (Preterm birth)) OR (IUGR)) OR (low Apgar score)) OR (abortion)) OR (miscarriage)) | 9/3/2023;54:29 | 168 |
| | (((((Obstetric Labor, Premature) OR "Obstetric Labor, Premature"[Mesh]) OR ((Premature Birth) OR "Premature Birth"[Mesh])) OR (((((((((((Survival of preterm birth) OR (Survival of preterm neonates)) OR (Survival of preterm babies)) OR (Survival of preterm infants)) OR (Outcome of preterm infants)) OR (Outcome of premature infants)) OR (Outcome of preterm neonates)) OR (Survival of premature birth)) OR (Survival of premature infant)) OR (Survival of premature neonates)) OR (Survival of premature babies))) AND ((((((((((Determinants of survival of preterm neonates) OR (Determinants of survival of preterm babies)) OR (Determinants of survival of preterm infants)) OR (Predictors of survival of preterm neonates)) OR (Predictors of survival of preterm infants)) OR (Predictors of survival of premature neonates)) OR (Predictors of survival of premature infants)) OR (Associated factors of preterm infants)) OR (Associated factors of preterm neonates)) OR (Associated factors of preterm babies)) | 24/12/2023;3:26PM | 5,611 |
| | **Specific** | | 282 |
| | **(((((Obstetric Labor, Premature) OR "Obstetric Labor, Premature"[Mesh]) OR ((Premature Birth) OR "Premature Birth"[Mesh])) OR (((((((((((Survival of preterm birth) OR (Survival of preterm neonates)) OR (Survival of preterm babies)) OR (Survival of preterm infants)) OR (Outcome of preterm infants)) OR (Outcome of premature infants)) OR (Outcome of preterm neonates)) OR (Survival of premature birth)) OR (Survival of premature infant)) OR (Survival of premature neonates)) OR (Survival of premature babies))) AND ((((((((((Determinants of survival of preterm neonates) OR (Determinants of survival of preterm babies)) OR (Determinants of survival of preterm infants)) OR (Predictors of survival of preterm neonates)) OR (Predictors of survival of preterm infants)) OR (Predictors of survival of premature neonates)) OR (Predictors of survival of premature infants)) OR (Associated factors of preterm infants)) OR (Associated factors of preterm neonates)) OR (Associated factors of preterm babies)) AND (Ethiopia)** | | |
| Google Scholar | Effect AND short inter pregnancy interval OR perinatal outcome AND short inter pregnancy interval AND maternal outcome OR effect short inter conception OR effect birth to conception OR unrecomended pregnancy interval OR short conception to birth OR child birth outcome OR pregnancy outcome AND SSA." | 11/3/2023-30/4/2023 | 427,073 |
| | Preterm neonates AND survival of preterm neonates OR survival of premature infants OR outcome of premature infants OR survival of preterm babies AND determinants of survival of premature infants OR survival of premature neonates OR survival of premature births OR survival of premature babies AND Ethiopia. | 24/12/2023 3:32 | 5,970 |
| From other databases | Effect of short inter pregnancy interval on perinatal and maternal outcome | 11/3/2023-30/4/2023 | 815 |
| | Survival of preterm neonates and its predictors in Ethiopia | | 675 |
| Total retrieved article | | | 454,983 |
| Number of included studies | | | 13 |

details imported into the JBI System for the Unified Management, Assessment and Review of Information (JBI SUMARI) (JBI, Adelaide, Australia) [30, 31]. The full texts of selected articles were assessed in detail against the inclusion criteria by two independent reviewers. Reasons for exclusion of papers at full text that do not meet the inclusion criteria was recorded and reported in the systematic review. Any disagreements that arise between the reviewers at each stage of the selection process resolved through discussion. The results of the search and the study inclusion process was reported in full in the final systematic review and presented in a Preferred Reporting Items for Systematic Reviews and Meta-analyses (PRISMA) flow diagram (Fig 1).

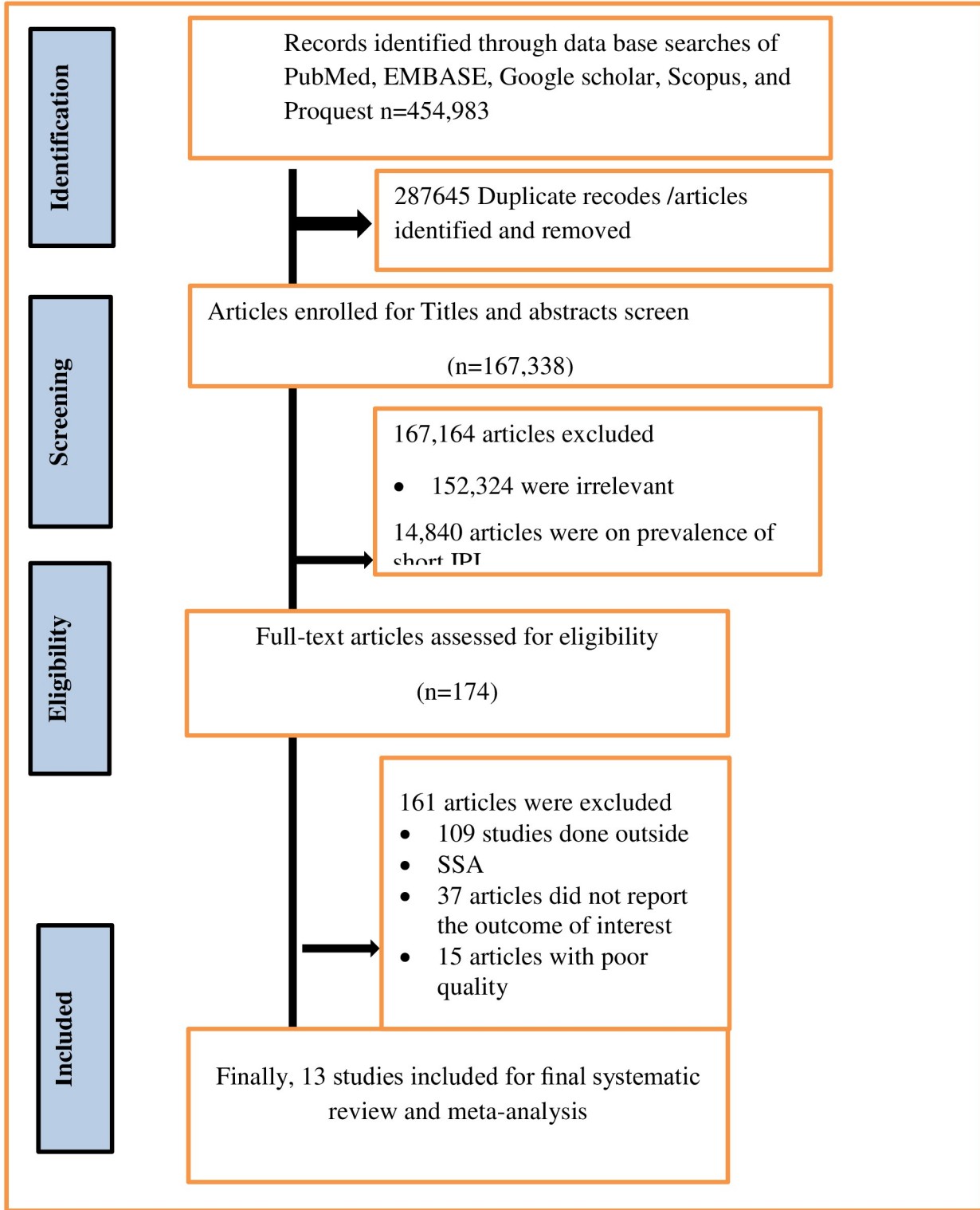

**Fig 1. PIRSMA flowchart diagram of the study selection.**

**Table 3. Quality assessment of included studies using the Joanna Briggs Institute criteria's for assessing quality of primary studies, 2019.**

| Study(Author) | Q1 | Q2 | Q3 | Q4 | Q5 | Q6 | Q7 | Q8 | Q9 | Score % |
|---|---|---|---|---|---|---|---|---|---|---|
| Larebo Y et al | Y | Y | Y | Y | N | Y | N | Y | Y | 77.78 |
| Ewnetu S et al | Y | Y | N | Y | N | Y | N | Y | Y | 66.67 |
| Nigatu B et al | Y | Y | Y | Y | Y | Y | Y | N | Y | 88.89 |
| Seyoum B et al | Y | Y | Y | Y | Y | Y | Y | N | Y | 88.89 |
| Dedecha W | Y | Y | N | Y | Y | Y | N | Y | Y | 77.78 |
| Muche A et al | Y | Y | Y | Y | Y | N | Y | Y | Y | 88.89 |
| Wolka E et al | Y | Y | Y | Y | Y | N | Y | N | Y | 77.78 |
| Atlaw D et al | Y | Y | Y | Y | Y | Y | Y | N | Y | 88.89 |
| Wakwoya E et al | Y | Y | N | Y | Y | Y | Y | Y | Y | 88.89 |
| Boda B et al | Y | Y | Y | Y | Y | Y | Y | N | Y | 88.89 |

Note:

Y—Yes, N—No, U—Unclear, NA- Not applicable

Q1 = Were the two groups similar and recruited from the same population?

Q2 = Were the exposures measured similarly to assign people to both exposed and unexposed groups?

Q3 = Was the exposure measured in a valid and reliable way?

Q4 = Were confounding factors identified?

Q5 = Were strategies to deal with confounding factors stated?

Q6 = Were the groups/participants free of the outcome at the start of the study (or at the moment of exposure)?

Q7 = Were the outcomes measured in a valid and reliable way?

Q8 = Was the follow up time reported and sufficient to be long enough for outcomes to occur?

Q9 = Was follow up complete, and if not, were the reasons to loss to follow up described and explored?

Q10 = Were strategies to address incomplete follow up utilized?

Q11 = Was appropriate statistical analysis used?

### Assessment of methodological quality

Eligible studies were critically appraised by two independent reviewers for methodological quality in the review using standardized critical appraisal instruments from JBI for the following studies: cohort, case control and analytical cross-sectional studies [28]. Any disagreements regarding quality appraisal that arose were resolved through discussion. All papers selected for quality appraisal were included in the review regardless of quality in order to be inclusive of all available evidence. Details of the quality appraisal are presented in (Tables 3–5).

### Data extraction

The extracted data from studies were included in the review by two independent reviewers using the standardized JBI data extraction tool available at JBI System for the Unified Management, Assessment and Review of Information (JBI SUMARI) (JBI, Adelaide, Australia) [30, 31]. The extracted data included specific details about the population, interventions, study methods, and outcomes of significance to the review question and its objectives. Any disagreements that arose between the reviewers were resolved through discussion or with a third reviewer.

### Data synthesis and statistical analysis

A meta-analysis was carried out to provide a comparative classification of the outcome and determinants of interest for the selected publications and to calculate the effect size for the

**Table 4. Quality assessment of included studies using the Joanna Briggs Institute criteria's for assessing quality of primary studies, 2019.**

| Study(Author) | Q1 | Q2 | Q3 | Q4 | Q5 | Q6 | Q7 | Q8 | Q9 | Score % |
|---|---|---|---|---|---|---|---|---|---|---|
| Larebo Y et al | Y | Y | Y | Y | N | Y | N | Y | Y | 77.78 |
| Ewnetu S et al | Y | Y | N | Y | N | Y | N | Y | Y | 66.67 |
| Nigatu B et al | Y | Y | Y | Y | Y | Y | Y | N | Y | 88.89 |
| Seyoum B et al | Y | Y | Y | Y | Y | Y | Y | N | Y | 88.89 |
| Dedecha W | Y | Y | N | Y | Y | Y | N | Y | Y | 77.78 |
| Muche A et al | Y | Y | Y | Y | Y | N | Y | Y | Y | 88.89 |
| Wolka E et al | Y | Y | Y | Y | Y | N | Y | N | Y | 77.78 |
| Atlaw D et al | Y | Y | Y | Y | Y | Y | Y | N | Y | 88.89 |
| Wakwoya E et al | Y | Y | N | Y | Y | Y | Y | Y | Y | 88.89 |
| Boda B et al | Y | Y | Y | Y | Y | Y | Y | N | Y | 88.89 |

Note:

Y—Yes, N—No, U–Unclear, NA- Not applicable

Q1 = Were the groups comparable other than the presence of disease in cases or the absence of disease in controls?

Q2 = Were cases and controls matched appropriately?

Q3 = Were the same criteria used for identification of cases and controls?

Q4 = Was exposure measured in a standard, valid and reliable way?

Q5 = Was exposure measured in the same way for cases and controls?

Q6 = Were confounding factors identified?

Q7 = Were strategies to deal with confounding factors stated?

Q8 = Were outcomes assessed in a standard, valid and reliable way for cases and controls

Q9 = Was the exposure period of interest long enough to be meaningful?

Q10 = as appropriate statistical analysis used?

**Table 5. Quality assessment of included studies using the Joanna Briggs Institute criteria's for assessing quality of primary studies, 2019.**

| Study(Author) | Q1 | Q2 | Q3 | Q4 | Q5 | Q6 | Q7 | Q8 | Q9 | Score % |
|---|---|---|---|---|---|---|---|---|---|---|
| Larebo Y et al | Y | Y | Y | Y | N | Y | N | Y | Y | 77.78 |
| Ewnetu S et al | Y | Y | N | Y | N | Y | N | Y | Y | 66.67 |
| Nigatu B et al | Y | Y | Y | Y | Y | Y | Y | N | Y | 88.89 |
| Seyoum B et al | Y | Y | Y | Y | Y | Y | Y | N | Y | 88.89 |
| Dedecha W | Y | Y | N | Y | Y | Y | N | Y | Y | 77.78 |
| Muche A et al | Y | Y | Y | Y | Y | N | Y | Y | Y | 88.89 |
| Wolka E et al | Y | Y | Y | Y | Y | N | Y | N | Y | 77.78 |
| Atlaw D et al | Y | Y | Y | Y | Y | Y | Y | N | Y | 88.89 |
| Wakwoya E et al | Y | Y | N | Y | Y | Y | Y | Y | Y | 88.89 |
| Boda B et al | Y | Y | Y | Y | Y | Y | Y | N | Y | 88.89 |

Note:

Y—Yes, N—No, U–Unclear, NA- Not applicable

Q1 = Were the criteria for inclusion in the sample clearly defined?

Q2 = Were the study subjects and the setting described in detail?

Q3 = Was the exposure measured in a valid and reliable way?

Q4 = Were objective, standard criteria used for measurement of the condition?

Q5 = Were confounding factors identified?

Q6 = Were strategies to deal with confounding factors stated?

Q7 = Were the outcomes measured in a valid and reliable way?

Q8 = Was appropriate statistical analysis used?

effect of short inter pregnancy interval on perinatal outcome in SSA. The related outcome of short inter pregnancy interval were examined based on eligibility requirements. With regard to one linked outcome of short inter pregnancy interval, at least two studies were taken into consideration, together with their respective measures of effect and 95% confidence intervals (CI). Calculating the effect size and 95% confidence interval provided an approximation of the substantial relationship between short inter pregnancy interval and its outcome (CI). A DerSimonian–Laird method-based random effects model was taken into consideration in order to identify variations both within and between studies [32]. In addition, $I^2$ statistics and Cochran's $Q$ test have been used to measure heterogeneity through studies. The percentage of the sample's overall variance that can be attributed to heterogeneity is thought to be measured by the $I^2$ statistics. $I^2$ values range from 0 to 100%, with $I^2 \geq$ **75%** signifying significant study heterogeneity [33]. We looked at publication bias qualitatively in the meta-analysis with funnel plot and used Begg's test and Egger's test ($P$ 0.05) to determine statistical significance [34]. STATA version 18 was used for the statistical analysis. The results are provided using texts, tables, and forest plots with measures of effect and 95% confidence interval.

### Heterogeneity and subgroup analysis

Using Galbraith plot analysis of the chosen studies, we investigate potential sources of heterogeneity. Galbraith plot was used to assess the impact of inappropriate studies. A Subgroup analysis was also performed by place of study and country.

Assessing certainty in the findings the summary of findings were created using GRADEPro GDT (McMaster University, ON, Canada). The summary of result was present the following information where appropriate: absolute risks for the intervention and control, estimates of relative risk, and a ranking of the quality of the evidence based on the risk of bias, directness, heterogeneity, precision and risk of publication bias of the review results. The outcomes reported in the summary of result were: low birth weight, preterm birth, PROM, IUGR, perinatal mortality and low APGAR score.

## Result

### Study selection

A total of 454,983researches were reviewed; after 287645 articles removed due to duplication; 167,338 studies were screened for titles and abstracts. There were 174 studies identified for full text retrieval; of these161 were not relevant and 13 studies were included in this review (Fig 1).

### Characteristics of the included studies

A total of 13 studies with 29,480 participants were considered. Of those, six studies [35–40] were conducted in Ethiopia, three studies [41–43] in Nigeria, two studies [44, 45] in Tanzania, and the rest two studies [46, 47] in other countries (Namibia and Sudan), respectively (Table 6).

### Perinatal and maternal outcome of short inter pregnancy interval

**Perinatal outcome.** *Low birth weight*. Nine of thirteen studies showed that low birth weight is statistically significant association with short inter pregnancy interval and the pooled effect also showed that significantly associated with (RR (95% CI) 1.98 (1.48, 2.47); $I^2$:62.97%. The heterogeneity test (P = 0.006) and $I^2$ = 62.97%showed that there is moderate heterogeneity/ variations across the studies. The result of Egger's test showed statistically significant publication bias (p = 0.001) (Fig 2).

**Table 6. Characteristics of studies which are included in the systematic review and meta-analysis, 2023.**

| No | Authors | Country | LBW AOR | Llb | Ulb | PTB AOR | LPT | UPT | IUGR AOR | Liu | Uiu | Apgar AOR | Lla | Ula | PROMAOR | Lpro | Upro | PM AOR | Lpm | Upm | Ane AOR | Lan | Uan |
|---|---|---|---|---|---|---|---|---|---|---|---|---|---|---|---|---|---|---|---|---|---|---|---|
| 1 | Gurmu et al | Ethiopia | 2.1 | 1.16 | 3.82 | 3.14 | 1.05 | 4.66 | 2.6 | 1.19 | 7.54 | 2.1 | 1.06 | 2.69 | 2.59 | 1.27 | 5.29 | | | | | | |
| 2 | Jenta et al | Ethiopia | 2.2 | 1.35 | 3.58 | 1.35 | 1.02 | 1.78 | | | | | | | | | | 3.83 | 1.9 | 7.71 | | | |
| 3 | Yesuf A et al | Ethiopia | 2.67 | 1.36 | 5.01 | 2.92 | 1.39 | 6.12 | | | | | | | | | | | | | | | |
| 4 | Brhane M et al | Ethiopia | | | | 6.85 | 3.07 | 15.31 | | | | | | | 2.96 | 1.53 | 5.72 | | | | | | |
| 5 | Jena B et al | Ethiopia | | | | | | | | | | | | | | | | 3.55 | 1.64 | 7.68 | | | |
| 6 | Korsa Et al | Ethiopia | 3.06 | 1.622 | 5.812 | | | | 3.78 | 2.36 | 6.09 | 4 | 2.594 | 6.166 | 2.16 | 1.688 | 4.702 | 6.65 | 3.046 | 14.514 | 3.97 | 2.994 | 5.269 |
| 7 | Henry C. Nnaji et al | Nigeria | 7.331 | 4.511 | 11.912 | | | | | | | | | | | | | | | | 7.759 | 4.779 | 12.599 |
| 8 | IFECHI O et al | Nigeria | | | | | | | | | | | | | | | | | | | 2.993 | 1.76 | 5.091 |
| 9 | Onwuka, et al | Nigeria | | | | | | | | | | | | | | | | | | | 2.09 | 1.44 | 3.03 |
| 10 | Mahande M et al | Tanzania | 1.61 | 1.34 | 1.71 | 1.52 | 1.31 | 1.74 | | | | | | | | | | 1.62 | 1.22 | 1.91 | | | |
| 11 | Lilungulu A et al | Tanzania | 6.7 | 3.6 | 12.3 | 9.78 | 4.90 | 19.5 | 7.7 | 3.80 | 15.74 | 6.9 | 3.6 | 13.1 | 13.6 | 7.2 | 25.64 | | | | 3.4 | 2.8 | 4.1 |
| 12 | Michael J et al | Namibia | 1.34 | 1.03 | 1.74 | 1.67 | 1.36 | 2.06 | | | | | | | | | | | | | 1.67 | 1.59 | 3.94 |
| 13 | Adam et al | Sudan | 1.9 | 1.001 | 3.5 | 2.3 | 1.10 | 4.7 | 2.6 | 1.19 | 7.54 | 2.1 | 1.06 | 2.69 | 2.1 | 1.27 | 5.29 | | | | | | |

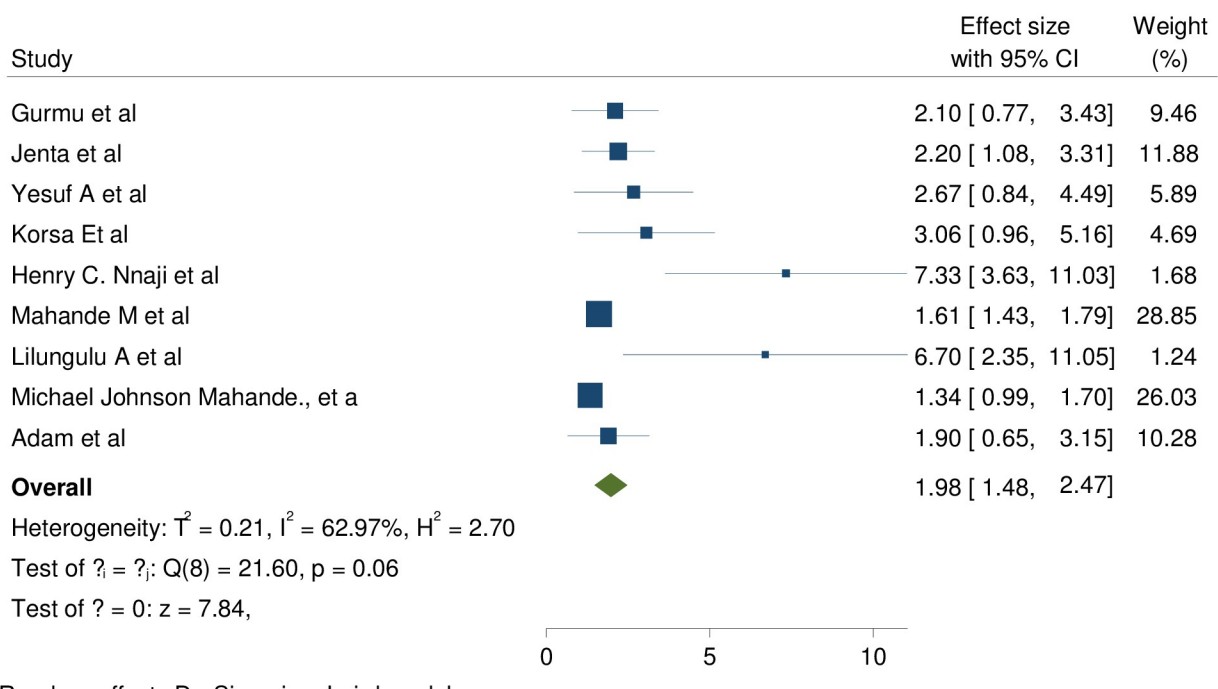

**Fig 2. Showing the effect of short inter pregnancy interval on birth weight.**

To minimize heterogeneity sub group analysis by study country was done and showed that those studies in Ethiopia were statistically significant with no evidence of variation across the studies with (RR (95% CI) 2.348(1.622, 3.074); $I^2$:0.0%; (P = 0.858) but other studies showed that high heterogeneity across the studies (RR (95% CI) 1.812 (1.181, 2.442); $I^2$: 76.06%; (P = 0.002) (Table 7).

## Preterm birth

Eight of thirteen studies report that preterm birth is statistically significant association with short inter pregnancy interval with pooled effect of (RR (95% CI) 1.67 (1.31, 2.03); $I^2$:51%. The heterogeneity test (P = 0.05) and $I^2$ = 51%showed that there is moderate heterogeneity/ variations across the studies. The result of Egger's test showed statistically significant publication bias (p = 0.0008) (Fig 3).

To minimize heterogeneity sub group analysis by study country was done and showed that in both groups there is heterogeneity across the studies (Table 8).

## Intrauterine growth retardation (IUGR)

Three studies evidenced that IUGR is statistically significant association with short inter pregnancy interval with pooled effect of (RR (95% CI) 3.78 (2.07, 5.49); $I^2$: 8.52%. The heterogeneity test (P = 0. 34) showed that no evidence of variation across studies. The result of Egger's test showed no statistically significant publication bias (p = 0.36) (Fig 4).

## Low APGAR score

Three studies verified that APGAR sore is statistically significant association with short inter pregnancy interval with pooled effect of (RR (95% CI) 3.49 (1.41, 5.57)); $I^2$: 71.11% %. The

**Table 7. Showing sub group analysis by study country of effect of short inter pregnancy interval on birth weight, 2023.**

| Study | Effect size | [95% conf. interval] | | % weight |
|---|---|---|---|---|
| **Group: Ethiopia** | | | | |
| Study 1 | 2.100 | 0.770 | 3.430 | 9.46 |
| Study 2 | 2.200 | 1.085 | 3.315 | 11.88 |
| Study 3 | 2.670 | 0.845 | 4.495 | 5.89 |
| Study 6 | 3.060 | 0.965 | 5.155 | 4.69 |
| Theta | 2.348 | 1.622 | 3.074 | |
| **Group: Nigeria+T~n|** | | | | |
| Study 7 | 7.331 | 3.630 | 11.032 | 1.68 |
| Study 10 | 1.610 | 1.425 | 1.795 | 28.85 |
| Study 11 | 6.700 | 2.350 | 11.050 | 1.24 |
| Study 12 | 1.340 | 0.985 | 1.695 | 26.03 |
| Study 13 | 1.900 | 0.650 | 3.149 | 10.28 |
| Theta | 1.812 | 1.181 | 2.442 | |
| **Overall** | | | | |
| Theta | 1.976 | 1.482 2.470 | | |

**Heterogeneity summary**

| Group | df | Q | P > Q | tau2 | % I2 | H2 |
|---|---|---|---|---|---|---|
| Ethiopia | 3 | 0.76 | 0.858 | 0.000 | 0.00 | 1.00 |
| Nigeria+Tanz~n | 4 | 16.71 | 0.002 | 0.241 | 76.06 | 4.18 |
| Overall | 8 | 21.60 | 0.006 | 0.211 | 62.97 | 2.70 |

Test of group differences: Q_b = chi2(1) = 1.19 Prob > Q_b = 0.

|  | | Effect size with 95% CI | Weight (%) |
|---|---|---|---|
| Study | | | |
| Gurmu et al | | 3.14 [ 1.33, 4.95] | 3.60 |
| Jenta et al | | 1.35 [ 0.97, 1.73] | 27.20 |
| Yesuf A et al | | 2.92 [ 0.55, 5.29] | 2.18 |
| Brhane M et al | | 6.85 [ 0.73, 12.97] | 0.34 |
| Mahande M et al | | 1.52 [ 1.30, 1.74] | 34.29 |
| Lilungulu A et al | | 9.78 [ 2.48, 17.08] | 0.24 |
| Michael Johnson Mahande., et a | | 1.67 [ 1.32, 2.02] | 28.52 |
| Adam et al | | 2.30 [ 0.50, 4.10] | 3.62 |
| **Overall** | | 1.67 [ 1.31, 2.03] | |

Heterogeneity: $T^2 = 0.09$, $I^2 = 51.00\%$, $H^2 = 2.04$

Test of $\theta_i = \theta_j$: Q(7) = 14.29, p = 0.05

Test of $\theta = 0$: z = 9.12

Random-effects DerSimonian–Laird model

**Fig 3. Showing the effect of short inter pregnancy interval on preterm birth.**

**Table 8. Showing sub group analysis by study country of effect of short inter pregnancy interval on preterm birth, 2023.**

| Study | Effect size | [95% conf. interval] | | % weight |
|---|---|---|---|---|
| **Group: Ethiopia** | | | | |
| Study 1 | 3.140 | 1.335 | 4.945 | 3.60 |
| Study 2 | 1.350 | 0.970 | 1.730 | 27.20 |
| Study 3 | 2.920 | 0.555 | 5.285 | 2.18 |
| Study 4 | 6.850 | 0.730 | 12.970 | 0.34 |
| Theta | 2.505 | 0.962 | 4.048 | |
| **Group: Nigeria+T~n\|** | | | | |
| Study 10 | 1.520 | 1.305 | 1.735 | 34.29 |
| Study 11 | 9.780 | 2.480 | 17.080 | 0.24 |
| Study 12 | 1.670 | 1.320 | 2.020 | 28.52 |
| Study 13 | 2.300 | 0.500 | 4.100 | 3.62 |
| Theta | 1.641 | 1.249 | 2.034 | |
| **Overall** | | | | |
| Theta | 1.672 | 1.312 | 2.031 | |
| **Heterogeneity summary** | | | | |

| Group | df | Q | P > Q | tau2 | % I2 | H2 |
|---|---|---|---|---|---|---|
| Ethiopia | 3 | 8.08 | 0.044 | 1.360 | 62.87 | 2.69 |
| Nigeria+Tanz~n | 3 | 6.01 | 0.111 | 0.063 | 50.09 | 2.00 |
| Overall | 7 | 14.29 | 0.046 | 0.086 | 51.00 | 2.04 |

Test of group differences: Q_b = chi2(1) = 1.13 Prob > Q_b = 0.288

heterogeneity test (P = 0.003) and $I^2$ = 71.11% showed that there is high heterogeneity/ variations across the studies. The result of Egger's test showed statistically significant publication bias (p = 0.01) (Fig 5).

## Premature Rupture of Membrane (PROM)

Four studies confirmed that PROM is statistically significant association with short inter pregnancy interval with pooled effect of (RR (95% CI) 2.87 (1.22, 4.51)); $I^2$: = 49.22% %. The

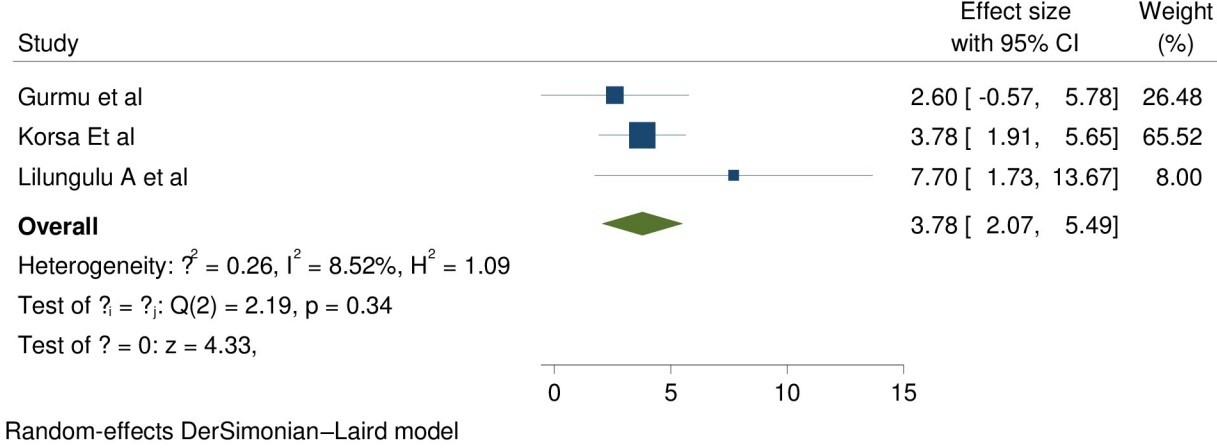

**Fig 4. Showing the effect of short inter pregnancy interval on IUGR.**

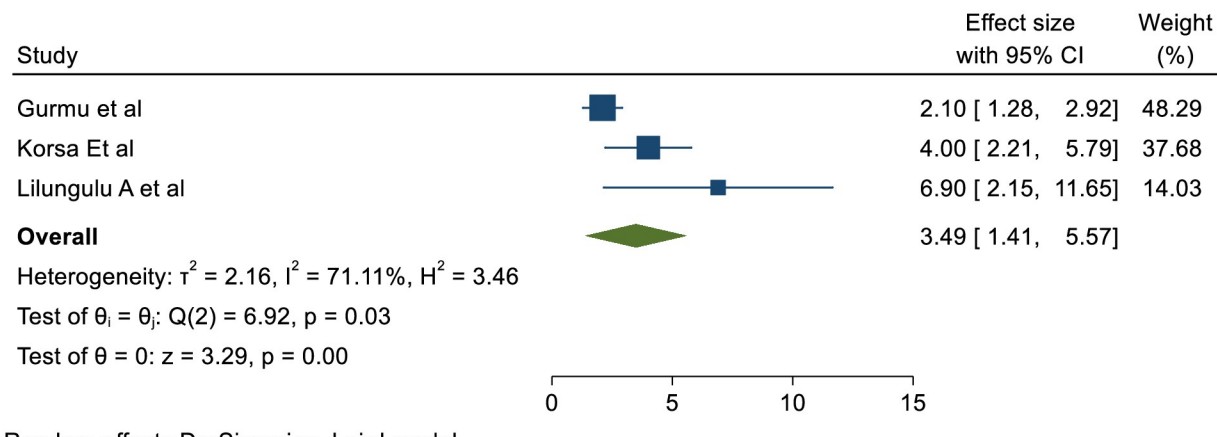

**Fig 5. Showing the effect of short inter pregnancy interval on APGAR score.**

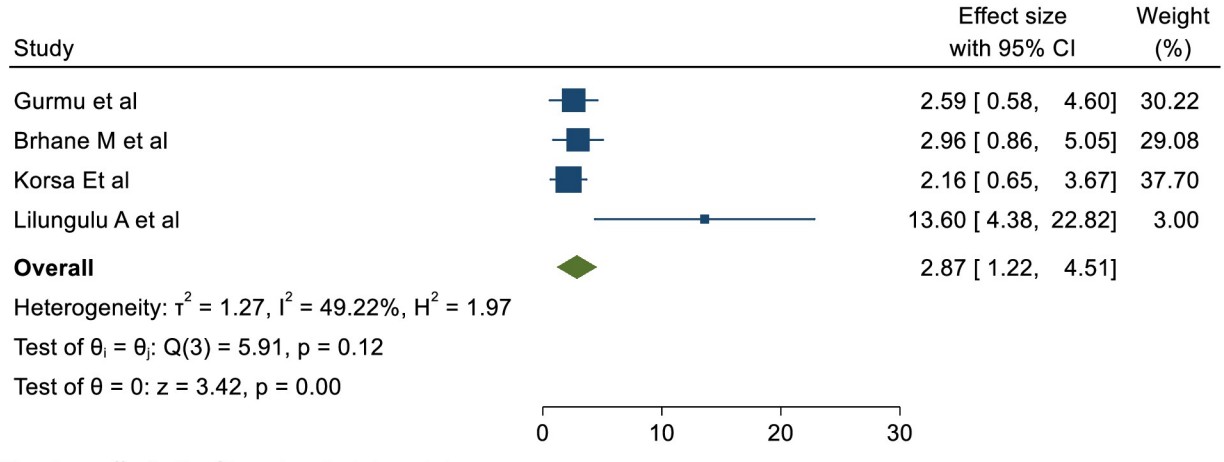

**Fig 6. Showing the effect of short inter pregnancy interval on PROM.**

heterogeneity test (P = 0.12) and I2 = 49.22%showed that there is moderate heterogeneity/ variations across the studies. The result of Egger's test showed statistically significant publication bias (p = 0.01) (Fig 6).

## Perinatal mortality

Four studies confirmed that PROM is statistically significant association with short inter pregnancy interval with pooled effect of (RR (95% CI) 2.95 (1.10, 4.81)); I2: = 54.37% %. The heterogeneity test (P = 0.09) and I2 = 54.37%showed that there is moderate heterogeneity/ variations across the studies. The result of Egger's test showed statistically significant publication bias (p = 0.01) (Fig 7).

## Maternal outcome

Six of thirteen studies showed that maternal anemia is statistically significant association with short inter pregnancy interval and the pooled effect also showed that significantly associated

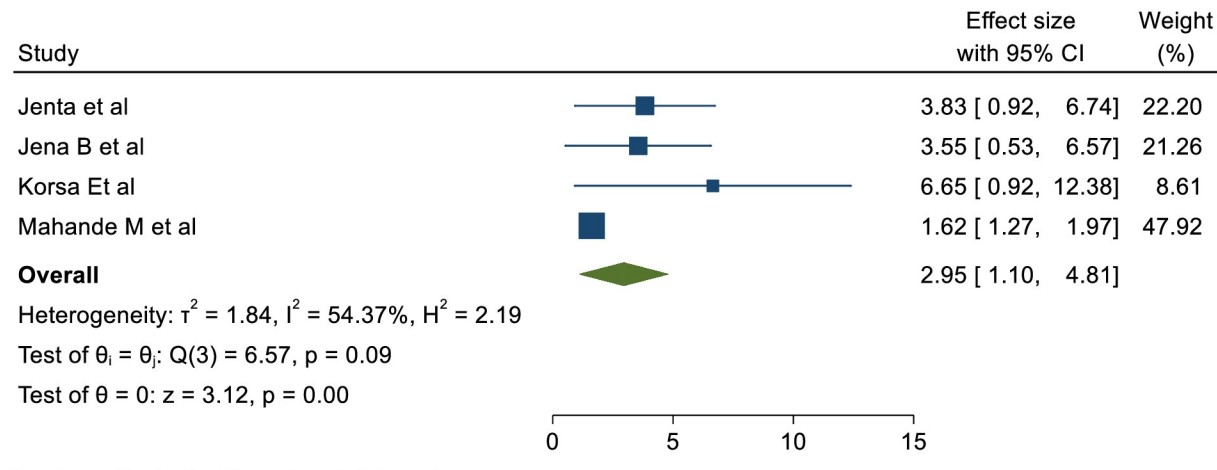

**Fig 7. Showing the effect of short inter pregnancy interval on perinatal mortality.**

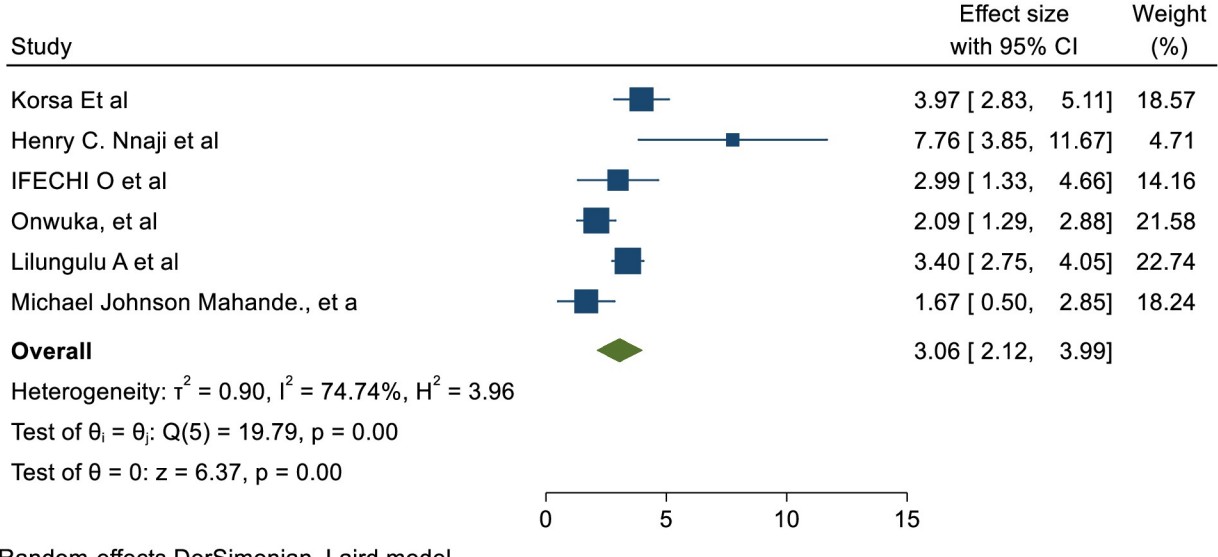

**Fig 8. Showing the effect of short inter pregnancy interval on maternal anemia.**

with (RR (95% CI) 3.06 (2.12, 3.99)); $I^2$: 74.74%. The heterogeneity test (P = 0.001) and $I^2$ = 74.74%showed that there is high heterogeneity/ variations across the studies. The result of Egger's test showed statistically significant publication bias (p = 0.05) (Fig 8).

## Discussion

This systematic review and meta-analysis assessed the effect of short inter pregnancy interval on perinatal and maternal outcomes. According to our review and meta-analysis finding; low birth weight, preterm birth, IUGR, low APGAR score, PROM and anemia were the main outcomes of short inter pregnancy interval.

In this review nine studies evidenced that women with SIPI highly prone to give low birth weight babies than women with optimal birth to pregnancy interval with pooled effect of 1.98 (1.48, 2.47); $I^2$:62.97%(; p = 0.006.The $I^2$ and P value showed that there is moderate heterogeneity across the studies. We were made subgroup analysis to minimize heterogeneity and showed that studies in Ethiopia were significantly associated without heterogeneity effect across the studies but in other group's showed significant heterogeneity. The heterogeneity might be due to methodological variations (designs and quality); Statistical heterogeneity, population and intervention variations. This study supported by studies in Latin America, Five Racial/Ethnic Groups in the United States, Lahore Pakistan, Guangdong Province China, Michigan, Utah, Northern Alberta &China [48–55]. This might be due to the following physiological and pathological changes;

Due to insufficient time to recover from the physiological stresses of the previous pregnancy before being subject to the stresses of the subsequent pregnancy, maternal nutritional depletion, which is a close succession of pregnancies and lactation periods, worsens the mother's nutritional status (To keep the requirements of the mother and the fetus in balance, a proper quantity of nutrients is necessary. The health of both the mother and the fetus will be in danger due to a state of biological rivalry brought on by a lack of supplies [23, 56, 57].

Increased risk of low birth weight may occur from the mother's nutritional state at conception being compromised and her capacity to support fetal growth being less than ideal.. Maternal nutritional depletion leads to maternal malnutrition fetal malnutrition and a compromised intrauterine environment, which would increase risk of low birth weight.

Others may be brought on by breastfeeding overlap; when pregnancies are closely spaced, breastfeeding and pregnancy overlap is more common, which may have an impact on how well the newborn nursed; changes in breastfeeding patterns or the composition and/or quantity of breast milk brought on by breastfeeding-pregnancy overlap may result in low birth weight [58–60].

In this review eight studies supported that women with SIPI gives preterm birth than women with optimal birth to pregnancy interval with pooled effect of 1.67 (1.31, 2.03); $I^2$:51%; p = 0.05.The $I^2$ and p value showed that there is moderate heterogeneity across the studies. The heterogeneity might be due to methodological variations (designs and quality); Statistical heterogeneity, population, intervention and outcome variations. This study reinforced by studies in Latin America, Five Racial/Ethnic Groups in the United States, Lahore Pakistan, Guangdong Province China, Michigan, Northern Alberta, Iraq, Scotland, California [48–51, 54, 55, 61]).

Short inter-pregnancy intervals may result in insufficient time for reproductive tissues to rebuild muscle tone after a pregnancy, which may increase the likelihood of cervical insufficiency at the end of the subsequent pregnancy and cause preterm birth [8, 62–64].

Short intervals between pregnancies may interfere with the normal processes of remodeling of endometrial blood vessels after delivery with subsequent utero placental under perfusion, increasing the risk of placental abruption, which ultimately results in preterm birth [9, 65].

Three studies evidenced that women with SIPI experienced more IUGR babies than women with optimal birth to pregnancy interval with pooled effect of 3.78 (2.07, 5.49); $I^2$: 8.52%; p: 0. 34. The value of $I^2$ and P showed that no evidence of variation across studies. This study strengthened by studies in Latin America, Guangdong Province China, Utah & Iraq [36, 39, 41, 51]. This due to the fact that short inter pregnancy interval results maternal nutritional depletion which results maternal malnutrition and the supply of nutrients from the mother to the fetus diminished finally results IUGR.

Three studies verified that newborns delivered from women with SIPI were lower APGAR score than women with optimal birth to pregnancy interval with pooled effect of 3.49 (1.41,

5.57)); $I^2$: 71.11% &p: 0.03. The Value of P and $I^2$ showed that there is high heterogeneity/ variations across the studies. The heterogeneity might be due to designs and quality variations; population, intervention and outcome variations. This is might be due to short inter pregnancy interval results maternal nutritional depletion results low birth weight and IUGR babies those factors might signifies low APGAR score at birth. And also short inter pregnancy interval results cervical incompetency and facilitates preterm birth; those preterm birth might be experienced low APGAR score at birth.

Four studies confirmed that women with SIPI were more prone to PROM than women with optimal birth to pregnancy interval with pooled effect of 2.87 (1.22, 4.51)); I2: = 49.22% and p: 0.12. The Value of $I^2$ showed that there is considerable heterogeneity/ variations across the studies. The heterogeneity might be due to designs and quality variations; population, intervention and outcome variations and smaller studies.

This may be because a brief inter-pregnancy period causes aberrant uterine blood vascular reorganization, which results in abruption placenta and may cause PROM. And as a result of maternal micronutrient deficiencies brought on by maternal nutrition deprivation. The alteration of collagen structure caused by micronutrient deficiencies that impact collagen synthesis has been linked to a higher risk of PPROM. [60, 61, 66, 67].

Four studies established that women with SIPI were more prone to perinatal mortality than women with optimal birth to pregnancy interval with pooled effect of) 2.95 (1.10, 4.81)); $I^2$: = 54.37% &p: 0.09. Even though $I^2$ showed that there is considerable heterogeneity but the heterogeneity test showed no significance. This study strengthened by studies in Latin America & British Columbia, Canada [36, 63].

Infectious illness transmission among siblings who live close to one another may be to blame for this, according to the Sibling Competition hypothesis. A family's young children may compete for resources, parental care, and attention if there are several young children there that are close in age [64].

The link between short inter-pregnancy intervals and newborn and child mortality may also be explained by the younger child being exposed to more infectious diseases, according to a different theory called transmission of infectious diseases among siblings. Due to the increased risk of exposure if more than one child of a vulnerable age is present in the household, children who are close in proximity would therefore be more likely to contract an infectious disease from one another [64].

Short gestational periods are linked to an increased risk of gastroenteritis, respiratory infections, and worm infestation, which can result in perinatal mortality [8, 65, 68].

Six of thirteen studies showed that women with SIPI were more exposes to anemia than women with optimal birth to pregnancy interval with pooled effect of 3.06 (2.12, 3.99)); I2: 74.74%&p: 0.001. This is might be pregnant women who do not take folic acid supplements, maternal serum and erythrocyte concentrations of folate begin to decline in the fifth month of pregnancy and continue to be low for several months after delivery. Short inter-pregnancy intervals cause insufficient replenishment of maternal folate resources. After delivery, breastfeeding mothers' maternal folate levels continue to be depleted. As lactation continues, the maternal tissue store is depleted as the amount of folate in breast milk rises. As a result, breastfeeding mothers may be more affected by short inter-pregnancy intervals on pregnancy outcomes, particularly if they don't replace their folate stores throughout the Interpregnancy interval or in the first trimester [9, 68]. Others might be due to nutritional depletion leads to macro and micro nutrient deficiency finally results nutritional deficiency anemia; And also due to frequent exposure of delivery process bleeding might results anemia secondary to bleeding.

## Limitations

Most of the finding of this review showed considerable heterogeneity, although we tried to perform subgroup analysis and meta-regression to find out the source of heterogeneity, we could not get the exact sources of heterogeneity. Due to limitation of published articles on effect of short inter pregnancy interval on perinatal and maternal outcome we consider only thirteen articles and this might be prone to heterogeneity across the studies.

## Conclusions

As per our review the main effect of short inter pregnancy interval is low birth weight, preterm birth, IUGR, low APGAR score, PROM, perinatal mortality and maternal anemia. This might be very useful for healthcare policymakers and NGOs to emphasize on it and discussing on the intervention strategies.

## Recommendations for practice

For the sake of maternal and child health, the World Health Organization (WHO) advised an ideal birth-to-pregnancy interval of at least 24 months or a birth-to-birth delay of 33 months or more in two subsequent deliveries. Taking the full advantage of this effort is recommended to maintain maternal and child health. Health care providers should practice accordingly to WHO recommendations. Healthcare policymakers and NGOs should ensure the application of this strategy and means of wide up policy and strategy to avoid short inter pregnancy interval.

## Recommendations for researcher

Further longitudinal and interventional studies needed to better quantify the impact of short inter pregnancy interval and to settle best strategies to overcome the problems.

## Acknowledgments

We would like to great thank to all authors of involved in the studies included in this systematic review and meta-analysis.

## Author Contributions

**Conceptualization:** Fentahun Yenealem Beyene.

**Data curation:** Fentahun Yenealem Beyene.

**Formal analysis:** Fentahun Yenealem Beyene.

**Funding acquisition:** Fentahun Yenealem Beyene.

**Investigation:** Fentahun Yenealem Beyene.

**Methodology:** Fentahun Yenealem Beyene, Kihinetu Gelaye Wudineh, Simachew Animen Bantie, Azimeraw Arega Tesfu.

**Project administration:** Fentahun Yenealem Beyene, Azimeraw Arega Tesfu.

**Resources:** Fentahun Yenealem Beyene, Simachew Animen Bantie, Azimeraw Arega Tesfu.

**Software:** Fentahun Yenealem Beyene, Simachew Animen Bantie, Azimeraw Arega Tesfu.

**Supervision:** Fentahun Yenealem Beyene, Simachew Animen Bantie.

**Validation:** Fentahun Yenealem Beyene, Kihinetu Gelaye Wudineh, Simachew Animen Bantie, Azimeraw Arega Tesfu.

**Visualization:** Fentahun Yenealem Beyene, Kihinetu Gelaye Wudineh, Simachew Animen Bantie.

**Writing – original draft:** Fentahun Yenealem Beyene, Kihinetu Gelaye Wudineh, Simachew Animen Bantie, Azimeraw Arega Tesfu.

**Writing – review & editing:** Fentahun Yenealem Beyene, Simachew Animen Bantie, Azimeraw Arega Tesfu.

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
