## [Decision Letter · Decision Letter 0]

4 Aug 2023

PONE-D-23-14404Effect of short inter-pregnancy interval on perinatal and maternal outcomes among pregnant women in SSA 2023 Systematic review and meta-analysisPLOS ONE

Dear Dr. Beyene,

Thank you for submitting your manuscript to PLOS ONE. After careful consideration, we feel that it has merit but does not fully meet PLOS ONE’s publication criteria as it currently stands. Therefore, we invite you to submit a revised version of the manuscript that addresses the points raised during the review process.

We look forward to receiving your revised manuscript.

Kind regards,

Ahmed Mohamed Maged, MD

Academic Editor

PLOS ONE

Journal Requirements:

2. Please amend the manuscript submission data (via Edit Submission) to include author "Azimeraw Arega Tesfu". 

3. Please amend your authorship list in your manuscript file to include author "Alemtsehay Mekonnen Munea".

5. We note you have included a table to which you do not refer in the text of your manuscript. Please ensure that you refer to Table 6 & 7 in your text; if accepted, production will need this reference to link the reader to the Table.

**Additional Editor Comments:**

Please respond to all reviewers comments

Reviewers' comments:

Reviewer's Responses to Questions

**Comments to the Author**

1. Is the manuscript technically sound, and do the data support the conclusions?

Reviewer #1: Yes

Reviewer #2: Yes

Reviewer #3: Yes

2. Has the statistical analysis been performed appropriately and rigorously? 

Reviewer #1: Yes

Reviewer #2: Yes

Reviewer #3: Yes

3. Have the authors made all data underlying the findings in their manuscript fully available?

Reviewer #1: Yes

Reviewer #2: Yes

Reviewer #3: Yes

4. Is the manuscript presented in an intelligible fashion and written in standard English?

Reviewer #1: No

Reviewer #2: Yes

Reviewer #3: Yes

5. Review Comments to the Author

Reviewer #1: abstract

overall very descriptive

introduction

please add more studies about your topic in introduction section

method

very good you need to justify in detailed your quality assessment and what criteria you used.

results

very good

Discussion

very good

Reviewer #2: 1. Figures references must be added.

2. There is no reference for most of the tables in the paper.

3. Unify the reference style.

4. Clarify future work.

5. Define all the abbreviations at the first appearance.

Reviewer #3: The work you sent is amazing, I must say after carefully reading it. The writers did a great job performing their research and clearly and succinctly presenting their findings. The paper contributes significantly to the body of knowledge by addressing a crucial problem in the field.

6. PLOS authors have the option to publish the peer review history of their article (what does this mean?). If published, this will include your full peer review and any attached files.

Reviewer #1: **Yes: **Ahmed Said Ali

Reviewer #2: No

Reviewer #3: No

---

## [Author Response · Author response to Decision Letter 0]

6 Nov 2023

To: Plose One Academic Editor 

PONE-D-23-14404 

From: Fentahun Yenealem

First of all I would like to say thank you very much for your timely response and your actively involvement on the comments and coordination. For national security reasons, I had to postpone submitting my thoughts. I appreciate your patience. And my heart full gratitude again reaches to academic editor and all reviewers that was giving a measurable, fruitful and observable comments and questions by devoting their golden time.

 Point by point response to Editor and Reviewer

I. Point by point response to Editor

Q#1. Please ensure that your manuscript meets PLOS ONE's style requirements, including those for file naming. The PLOS ONE style templates can be found at 

Response 1: We accept the recommendations and we have reviewed based on the formats.

Q#2. Please amend the manuscript submission data (via Edit Submission) to include author "Azimeraw Arega Tesfu"

Response 2: We accept the comments and recommendations and we have added Azimeraw Arega Tesfu to the submission data.

Q#3. Please amend your authorship list in your manuscript file to include author "Alemtsehay Mekonnen Munea".

Response 3: We accept the comments and recommendations and we have deleted Alemtsehay Mekonnen Munea" from the submission data; since she has no any contribution in this paper; it is typing error on submission , to write Azimeraw Arega Tesfu we wrote Alemtsehay Mekonnen Munea" 

Q#4. Please include a separate caption for each figure in your manuscript.

Response 4: We accept the comments and recommendations and we have incorporated a separate caption for each figure in the manuscript.

Q# 5. We note you have included a table to which you do not refer in the text of your manuscript. Please ensure that you refer to Table 6 & 7 in your text; if accepted, production will need this reference to link the reader to the Table.

Response 5: We accept the comments and recommendations and we have referred each table and figure based on the journal templates. We have referred table 6&7 in result section line 144 and 157 respectively. 

Q# 6. Please include captions for your Supporting Information files at the end of your manuscript, and update any in-text citations to match accordingly. Please see our Supporting Information guidelines for more information: http://journals.plos.org/plosone/s/supporting-information. 

Response 6: We accept the comments and recommendations and we have included captions of Supporting Information files at the end of the manuscript based on the journal templates.

Q#7. Please review your reference list to ensure that it is complete and correct. If you have cited papers that have been retracted, please include the rationale for doing so in the manuscript text, or remove these references and replace them with relevant current references. Any changes to the reference list should be mentioned in the rebuttal letter that accompanies your revised manuscript. If you need to cite a retracted article, indicate the article’s retracted status in the References list and also include a citation and full reference for the retraction notice.

Response 7: We accept the recommendations and we have reviewed and all reference lists complete and correct.

II. Point by point response to Reviewers 

For Reviewer one 

Q#1. Please add more studies about your topic in introduction section

Response 1: We accept the recommendations and we had explored more and more as our knowledge.

Q#2. Method section very good you need to justify in detailed your quality assessment and what criteria you used

Response 2: Thank you for your recommendations and we had tried to explore in the quality assessment section “line 96”.

For Reviewer two 

Q#1. Figures references must be added.

Response 1: Thank you for your recommendations; do you main captions” we have incorporated all Figures captions. Reference for figures needed if you use directly others result. 

Q#2. There is no reference for most of the tables in the paper.

Response 2: Thank you for your comments and recommendations; do you main captions” we have incorporated all table captions. Reference for tables needed if you use directly others result. 

Q#3 Unify the reference style.

Response 3: Thank you for your comments and recommendations; And we have cited the references based on journal guide lines. 

Q#4. Clarify future work.

Response #4: Thank you for your comments and recommendations; and we have put our recommendations / future work/ on line 311-318.

Q#5.Define all the abbreviations at the first appearance.

Response 5: Thank you for your comments and recommendations; And we have put the abbreviations based on journal guide lines; next to the recommendation section.

For Reviewer three

Thank you for your reading and giving me appreciation.

 Thank you!!!

Fentahun Yenealem

---

## [Editor Report · Decision Letter 1]

8 Nov 2023

Effect of short inter-pregnancy interval on perinatal and maternal outcomes among pregnant women in SSA 2023:  Systematic review and meta-analysis

PONE-D-23-14404R1

Dear Dr. Beyene,

We’re pleased to inform you that your manuscript has been judged scientifically suitable for publication and will be formally accepted for publication once it meets all outstanding technical requirements.

Kind regards,

Ahmed Mohamed Maged, MD

Academic Editor

PLOS ONE
---

## [Editor Report · Acceptance letter]

21 Jun 2024

PONE-D-23-14404R1 

PLOS ONE

Dear Dr. Beyene, 

I'm pleased to inform you that your manuscript has been deemed suitable for publication in PLOS ONE. Congratulations! Your manuscript is now being handed over to our production team.

Kind regards, 

on behalf of

Professor Ahmed Mohamed Maged 

Academic Editor

PLOS ONE